# Comparison of motif-based and whole-unique-sequence-based analyses of phage display library datasets generated by biopanning of anti-*Borrelia burgdorferi* immune sera

Yurij Ionov[1]*, Artem S. Rogovskyy[2]*

**1** Department of Cancer Genetics, Roswell Park Comprehensive Cancer Center, Buffalo, New York, United States of America, **2** Department of Veterinary Pathobiology, College of Veterinary Medicine and Biomedical Sciences, Texas A&M University, College Station, Texas, United States of America

* Yurij.Ionov@RoswellPark.org (YI); arogovskyy@tamu.edu (ASR)

**Data Availability Statement:** All relevant data are within the paper and its Supporting Information files.

## Abstract

Detection of protection-associated epitopes via reverse vaccinology is the first step for development of subunit vaccines against microbial pathogens. Mapping subunit vaccine targets requires high throughput methods, which would allow delineation of epitopes recognized by protective antibodies on a large scale. Phage displayed random peptide library coupled to Next Generation Sequencing (PDRPL/NGS) is the universal platform that enables high-yield identification of peptides that mimic epitopes (mimotopes). Despite being unsurpassed as a tool for discovery of polyclonal serum mimotopes, the PDRPL/NGS is far inferior as a quantitative method of immune response. Difficult-to-control fluctuations in amounts of antibody-bound phages after rounds of selection and amplification diminish the quantitative capacity of the PDRPL/NGS. In an attempt to improve the accuracy of the PDRPL/NGS method, we compared the discriminating capacity of two approaches for PDRPL/NGS data analysis. The whole-unique-sequence-based analysis (WUSA) involved generation of 7-mer peptide profiles and comparison of the numbers of sequencing reads for unique peptide sequences between serum samples. The motif-based analysis (MA) included identification of 4-mer consensus motifs unifying unique 7-mer sequences and comparison of motifs between serum samples. The motif comparison was based not on the numbers of sequencing reads, but on the numbers of distinct 7-mers constituting the motifs. Our PDRPL/NGS datasets generated from biopanning of protective and non-protective anti-*Borrelia burgdorferi* sera of New Zealand rabbits were used to contrast the two approaches. As a result, the principle component analyses (PCA) showed that the discriminating powers of the WUSA and MA were similar. In contrast, the unsupervised hierarchical clustering obtained via the MA classified the preimmune, non-protective, and protective sera better than the WUSA-based clustering. Also, a total number of discriminating motifs was higher than that of discriminating 7-mers. In sum, our results indicate that MA approach improves the accuracy and quantitative capacity of the PDRPL/NGS method.

**Funding:** The authors received no specific funding for this work.

**Competing interests:** The authors have declared that no competing interests exist.

**Abbreviations:** BETA, Visualizing clustering of multivariate data; IACUC, Institutional Animal Care and Use Committee of Texas A&M University; NGS, Next Generation Sequencing technology; NZW, New Zealand White rabbits; PCA, Principal Component Analysis; PDRPL, Random peptide phage display library.

## Introduction

Identification of protection-associated (PA) antigens and/or their epitopes via reverse vaccinology is one of the very first steps for development of vaccines against human pathogens [1–9]. To select the most promising vaccine candidate(s) initially requires mapping multiple PA targets, which then can be individually tested for their protective efficacy via animal immunization assays [10]. To map numerous PA epitopes, high throughput methods, which would allow PA targets to be specifically recognized by protective serum antibodies on a large scale, are needed.

The high-density peptide microarray is a traditional method for mapping linear epitopes of antibodies developed against protein antigens [11–13]. The microarray contains overlapping peptides that cover the entire length of a target protein. However, this relatively straightforward method is not optimal for analyzing repertoires of antibodies developed against intact bacterial pathogens because these microorganisms are represented by an enormously complex mixture of antigens. Generation and application of customized microarrays with peptides that would encompass an entire proteome of a virus or bacterium is highly cost-prohibitive and time-consuming.

An alternative to customized platforms is the random peptide array, where $10^4$ 17-mer random peptide sequences selected by a random number generator are printed on glass slides [14]. When a serum sample is applied to the surface of this peptide array, distinct binding patterns of the unique molecular recognition elements associated with pathogen-specific antibodies are generated. The random peptide array is highly reproducible and, hence, allows changes in the antibody repertoires that are characteristic for different sates of a disease to be detected [15, 16]. However, the disadvantage is that densely surface-immobilized ligands can amplify non-specific interactions of low affinity, which reduces the overall capacity of the random peptide array for identifying disease-specific interactions. The other drawback is that the total number of peptides that can be attached to a single 25×75-mm glass is limited. Despite previous studies successfully used $10^4$ random peptides to characterize different immune responses [17], the assay with such a low number of ligands can still miss important antibody reactivities.

Phage displayed random peptide library (referred to here as PDRPL) is an inexpensive alternative to the high-density peptide array. The first application of PDRPL demonstrated the feasibility of affinity selection for identifying a peptide mimetic, which represented cognate epitopes of monoclonal antibodies [18]. Since then, PDRPL has been extensively applied to characterize specificity of polyclonal sera from cancer or infectious disease patients [19–24]. Similar to the random peptide array, PDRPL is also a universal platform, yet with a much higher capacity to represent sequences of a complete proteome of any microorganism. For example, a commercially available Ph.D.-7 library of random heptapeptides (New England Biolabs, USA) has a complexity on the order of $10^9$ independent clones and thus can represent any unique 7-mer peptide sequence (referred to here as 7-mer sequence). According to the product specifications, the phage concentration of the Ph.D.-7 library is $10^{13}$ phage particles per mL. The latter means that 10 μl of the library used for a biopanning experiment contains $10^{11}$ phage particles and each 7-mer is thus represented on average by almost 80 times (which is calculated by dividing $10^{11}$ by $20^7$). In addition to linear epitopes, phage displayed peptides may also represent both linear and conformational epitope mimetics of protein and carbohydrate antigens also known as mimotopes [25].

Until a recent advent of Next Generation Sequencing technology (referred to here as NGS), the main disadvantage of traditional PDRPL was the necessity to individually sequence phage clones. The NGS has transformed the phage display into a high throughput method, which, to date, allows the diversity of antibody specificities to be thoroughly examined [26]. By using the

NGS, hundred thousands of affinity-selected peptide sequences can be obtained in a single sequencing run [27]. As a result, the PDRPL coupled with the NGS (referred to here as PDRPL/NGS) has been extensively utilized to analyze repertoires of serum antibodies [28–30].

In our recent studies [10, 31, 32], we successfully applied the PDRPL/NGS to define repertoires of antibodies developed in mice and rabbits that had an active infection with *Borrelia burgdorferi*, the bacterial agent of Lyme disease (LD) [33–41]. LD pathogen has the capacity to establish persistent infection in mice, the primary natural mammalian reservoir of *B. burgdorferi* in the United States [42–47], and human patients [48–53], despite very strong anti-*B. burgdorferi* antibody responses [54–60]. In immunocompetent mouse models, the ability of *B. burgdorferi* to successfully evade otherwise potent antibodies is mainly attributed to the highly efficacious VlsE-encoding system, whose genetic removal results in rapid clearance of LD pathogen by host antibodies [54, 59, 61–69]. In contrast to experimental mouse or any other animal LD models [70–86], however, New Zealand White (NZW) rabbits develop a protective antibody-mediated response, which effectively clears *B. burgdorferi* within 3–8 weeks, despite the antigenically varying surface protein, VlsE [32, 87, 88]. Our recent study has examined repertoires of antibodies in sera collected from *B. burgdorferi*-infected NZW rabbits at day 14 and 28 postinfection, the time points at which the rabbit antibodies were shown to be non-protective or protective against the LD pathogen in mice, respectively [32].

In the present work, we used our previously generated PDRPL/NGS data to improve on the PDRPL/NGS analysis in discriminating between the protective and non-protective anti-*B. burgdorferi* sera of NZW rabbits [32]. Specifically, in order to extract more information on any qualitative and quantitative alterations in anti-*B. burgdorferi* antibody specificities, we have compared two distinct modes of analyses of the PDRPL/NGS data. The whole-unique-sequence-based analysis (referred to here as WUSA) involved 7-mer sequences and compared the number of sequencing reads for each unique peptide between the protective and non-protective sera after the normalization for the total number of all the sequences. In contrast, the alternative motif-based (referred to here as MA) method included identification and comparison of four-amino-acid-long consensus motifs (referred to here as 4-mer motifs), each representing a cluster of related 7-mer sequences that differed by only few amino acids.

## Results

Our previous study has demonstrated that by day 28 postinfection, NZW rabbits developed a repertoire of protective antibodies against *B. burgdorferi* infection [32]. Specifically, it was shown that, when passively transferred to naïve mice, the rabbit antibodies prevented infection by highly immune-evasive *B. burgdorferi*. Moreover, in mice with ongoing infection, the rabbit antibodies were able to significantly reduce LD-induced inflammation in joints [32], the avascular collagenous tissues to which antibody access is somewhat limited [89]. Our follow-up study defined specificities of day-14 (non-protective) and day-28 (protective) rabbit antibodies via the PDRPL/NGS and then directly compared the two repertoires. As a result, specificities of the rabbit protective antibodies and their respective targets were identified [9].

In the present study, we have compared two distinct approaches, the WUSA and MA, for the analysis of the PDRPL/NGS data [9]. Both approaches directly compared mimotope sequences between the day-14 (non-protective) and day-28 (protective) immune serum samples from 3 NZW rabbits (animals P, Y, and Z). The preimmune sera were pooled from the 3 rabbits prior to their challenge with wild-type *B. burgdorferi* B31-A3 strain and served as a background control. To obtain mimotope sequences, 1 pooled preimmune, 3 non-protective, and 3 protective serum samples (referred to here as PI, NP, and PR samples, respectively) were analyzed via commercially available Ph.D.-7 library of random heptapeptides followed by

sequencing via Illumina 2500 sequencing platform [9]. One PR sample was analyzed in duplicate (samples P28a and P28b). For both methods, 1,000 most abundant 7-mer sequences generated from the PDRPL/NGS analysis of NP (P14, Y14, and Z14), PR (P28a, P28b, Y28, and Z28), and PI samples were selected.

## Analyzing the protective and non-protective rabbit serum samples via the MA approach

In the MA approach, we first identified mimotope motifs that represented clusters of 7-mer sequences. Within each cluster, the respective 7-mers differed by few amino acids outside their 4-mer motif. Motifs, in turn, could have conservative amino acid substitutions. To generate 50 4-mer motifs for each sample, a total of 8 mimotope sequence datasets (P14, Y14, Z14, P28a, P28b, Y28, Z28, and PI) were separately processed via MEME software tool (http://meme-suite.org/tools/meme) [90]. Each dataset consisted of 1,000 7-mers and each motif was derived from at least 3 distinct sequence variants. To make the 7-mer sequences compatible with the MEME tool (the minimum length is 8 letters), the letter X was added to each 7-mer. Prior to running the program, we chose that each motif has the minimum sites of 3 (to unify at least 3 unique sequences) and the width of 4 amino acids. The MEME output exemplified in Fig 1 shows two abundant motifs, QKPL and KIGD, which unified 9 and 46 distinct 7-mer sequences, respectively, for sample Z28. As a result of this *in silico* analysis, we found that

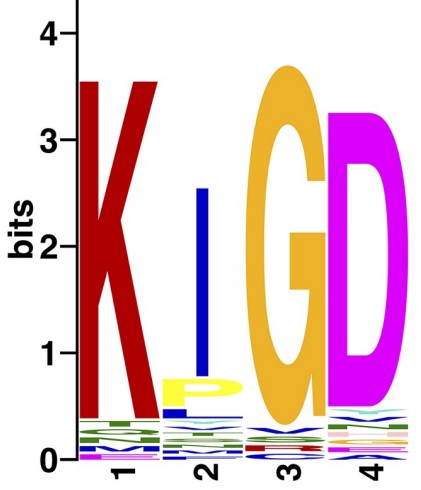

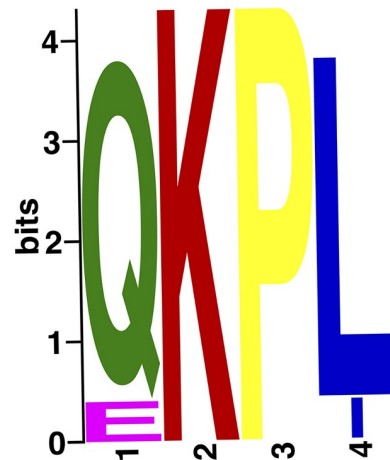

**Log Likelihood Ratio:** 443 [?]   **Information Content:** 13.1
**Relative Entropy:** 13.9 [?]   **Bayes Threshold:** 6.37562 [?]

| Name [?] | Start [?] | p-value [?] | Sites [?] |
|---|---|---|---|
| 900. 900 | 3 | 3.16e-6 | QV **KIGD** KX |
| 811. 811 | 3 | 3.16e-6 | YD **KIGD** KX |
| 741. 741 | 4 | 3.16e-6 | FPL **KIGD** X |
| 732. 732 | 3 | 3.16e-6 | YI **KIGD** KX |
| 647. 647 | 3 | 3.16e-6 | YV **KIGD** QX |
| 511. 511 | 3 | 3.16e-6 | YA **KIGD** KX |
| 471. 471 | 3 | 3.16e-6 | YF **KIGD** KX |
| 454. 454 | 3 | 3.16e-6 | YV **KIGD** TX |
| 390. 390 | 3 | 3.16e-6 | YV **KIGD** NX |

**Log Likelihood Ratio:** 96 [?]   **Information Content:** 16.3
**Relative Entropy:** 15.4 [?]   **Bayes Threshold:** 8.4507 [?]

| Name [?] | Start [?] | p-value [?] | Sites [?] |
|---|---|---|---|
| 860. 860 | 4 | 1.38e-5 | RLL **QKPL** X |
| 780. 780 | 4 | 1.38e-5 | GLV **QKPL** X |
| 757. 757 | 4 | 1.38e-5 | GRL **QKPL** X |
| 659. 659 | 4 | 1.38e-5 | DLL **QKPL** X |
| 590. 590 | 4 | 1.38e-5 | ALL **QKPL** X |
| 529. 529 | 4 | 1.38e-5 | SLL **QKPL** X |
| 153. 153 | 4 | 1.38e-5 | VLL **QKPL** X |
| 220. 220 | 4 | 1.89e-5 | NAA **QKPI** X |
| 858. 858 | 4 | 2.72e-5 | GLL **EKPL** X |

**Fig 1. Examples of MEME-generated 4-mer motifs unifying the unique 7-mer peptide sequences.**

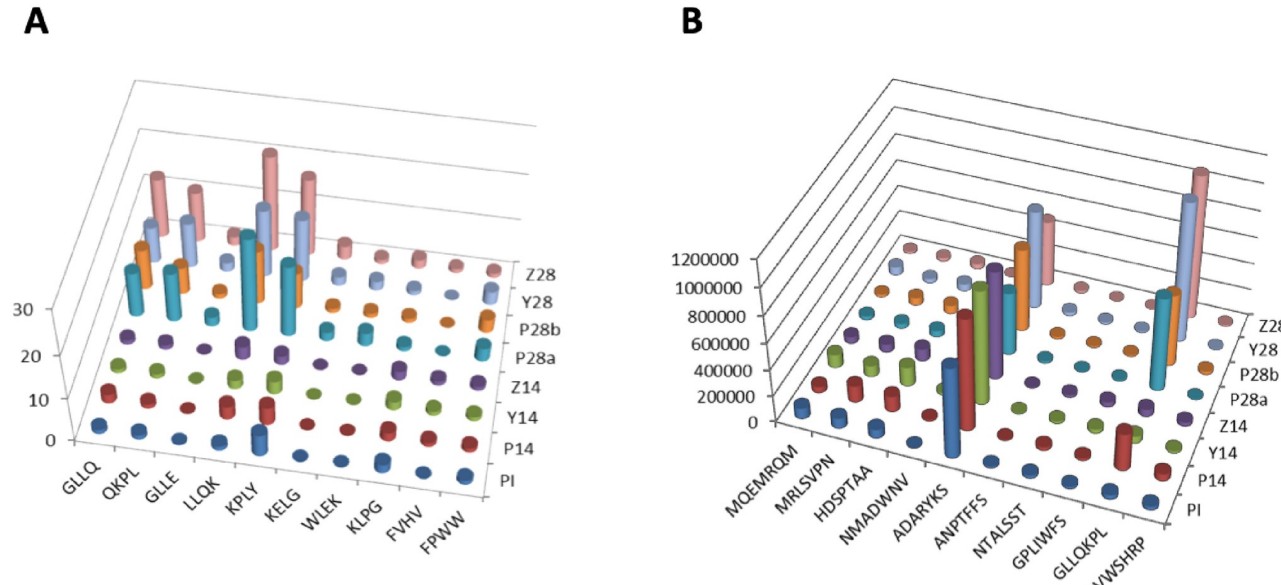

**Fig 2. Frequency distribution across rabbit serum samples for the discriminating 4-mer motifs (A) and unique 7-mer peptides (B) with the lowest *p*-values according to the *t*-test.**

some motifs were shared by the 8 datasets. After the redundant motifs were removed, there was a total of 330 distinct motifs for the 8 samples. S1 Table provides a complete list of these motifs and the respective numbers of the 7-mer sequences that constituted each motif for each serum sample. After performing the paired two-tailed *t*-test (leaving out one of the two technical replicates for one protective sample), we identified 9 most significant motifs that discriminated between the NP and PR samples ($p<0.1$). Furthermore, out of these 9 motifs, 4 motifs were most significant ($p<0.05$) and were derived from a most frequently represented 7-mer for the 4 PR samples, GLLQKPL. Fig 2A shows the distribution of 10 selected motifs that were most significant for the 8 samples. Since the *t*-test may be not the best statistics to identify discriminating motifs, a different method was also tested. For that, the minimum number of motif variants for the PR samples was compared with the maximum number of motif variants for the NP samples. As a result, we identified a total of 10 motifs, whose minimum numbers for the PR samples were higher than the maximum numbers for the NP samples. Interestinlgy, 7 of these 10 motifs were identical to the most statistically significant motifs identified via the *t*-test (S1 Table, Sheet 2).

We then performed Principal Component Analysis (PCA) on the 330 motifs by using the web tool, BETA, which was developed to visualize clustering of multivariate data (https://biit.cs.ut.ee/clustvis/) [91]. Since the first principle component, which determines the direction of highest variability in the data, captured only 18% of the variability, the plot showed a high degree of data randomness (Fig 3A). Despite the observed randomness, however, the PCA analysis still separated the 4 PR samples from all the 3 NP and 1 PI samples (Fig 3A). The heatmap generated by the unsupervised hierarchical clustering via the ClustVis also demonstrated a high degree of variability between the 8 samples (Fig 3B). However, similar to the PCA plot, the respective heatmap showed that the PI sample was well segregated from the 4 PR samples; and yet neighbored with a cluster of the 3 NP samples (Fig 3B). Consistently, the duplicate samples, P28a and P28b, were grouped together despite a high degree of variability within each technical replicate (Fig 3B). Of note, the proximity of PCA values for samples Z14 and Z28 and

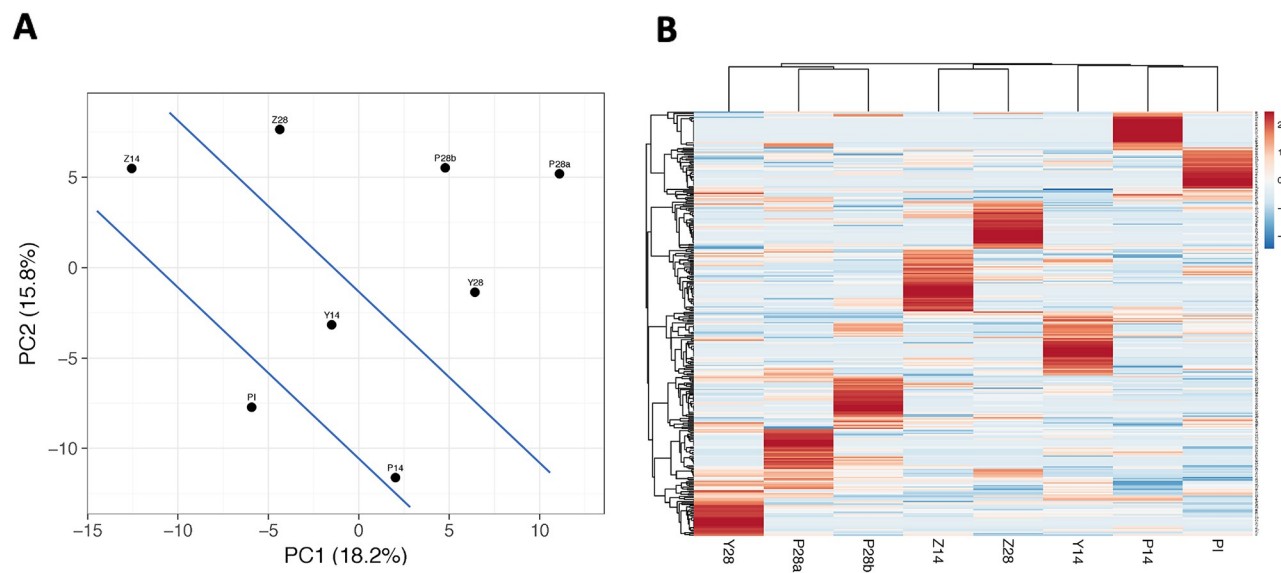

**Fig 3. Principal Component Analysis (PCA) plot (A) and heatmap (B) generated, via the ClustVis, by the motif-based analysis (MA).** PDRPL/NGS data were produced by biopanning of preimmune (PI), non-protective (P14, Y14, and Z14), and protective (P28a, P28b, Y28, and Z28) rabbit serum samples. Blue lines drawn across the PCA plot (A) separate the 3 groups of serum samples.

their respective heatmap patterns suggested that NZW rabbit Z had developed a protective antibody repertoire later compared to animals P and Y. Indeed, in contrast to NZW rabbits P and Y, skin biopsies of animal Z sampled at day 28 postinfection were only weakly culture-positive for *B. burgdorferi*, whose very low spirochetal numbers and impaired (slow) motility indicated a relatively late clearance [32].

## Analyzing the protective and non-protective rabbit serum samples via the WUSA approach

To compare the discriminating capacity of the MA with that of the more conventional WUSA method, the same 8 mimotope sequence datasets (P14, Y14, Z14, P28a, P28b, Y28, Z28, and PI) were analyzed via the WUSA. First, we calculated the number of sequencing reads for each 7-mer sequence, which reflected the relative titers of the respective mimotope-recognizing antibodies (1S Table). Thus, the numbers of sequencing reads were used to directly compare levels of antibodies of distinct specificities within and between the serum samples. Importantly, the numbers of 7-mers for each sample were normalized to have totals of sequencing reads equal across the 8 samples. This normalization was necessary to eliminate any biases that are associated with potential differences in amplification efficiency between the PCR reactions performed prior to multiplexing a phage DNA library.

For each dataset, we selected the top 50 most abundant sequences to match with the number of motifs (n = 50) used for the MA method. After eliminating the redundant sequences, there was a total of 172 distinct 7-mers for the 8 datasets. S2 Table lists the unique 7-mer sequences and the respective numbers of their sequencing reads. After performing the paired two-tailed *t*-test, we identified 11 most significant 7-mers that discriminated between the NP and PR samples ($p<0.1$). Fig 2B shows the 10 peptide sequences that had the lowest *p*-values. Then, the minimum numbers of sequencing reads for the PR samples were compared with the respective maximum values for the NP samples. Consequenlty, only 3 out of 11 peptides ($\alpha = 0.1$) had the higher numbers of their sequencing reads for the PR samples compared to the NP

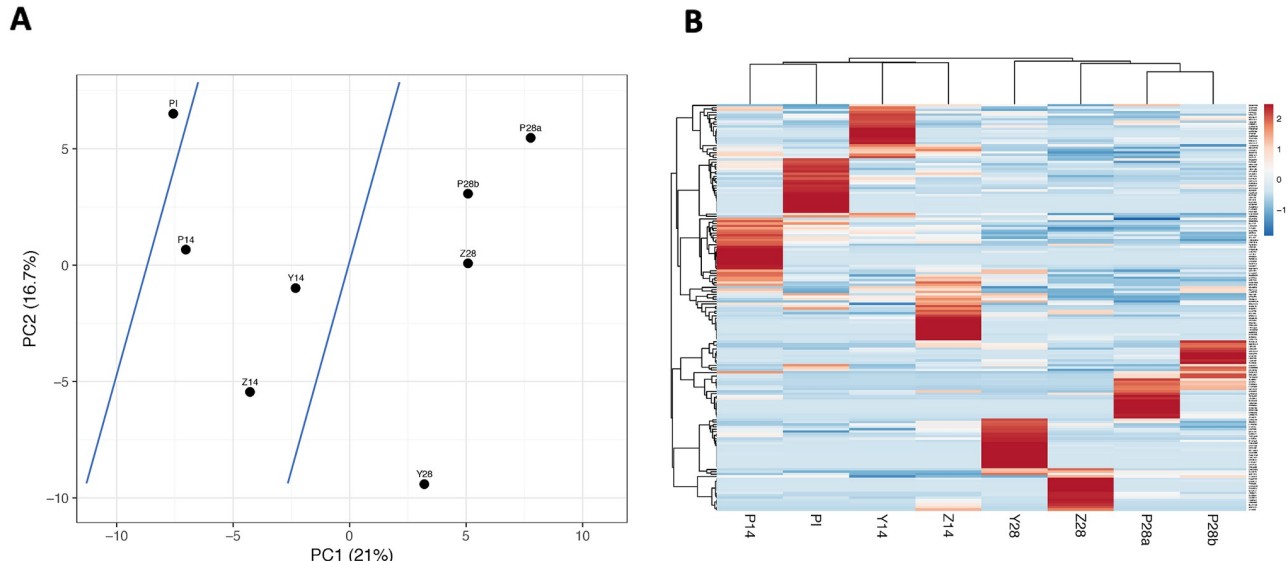

**Fig 4. Principal Component Analysis plot (A) and heatmap (B) generated, via the ClustVis, by the whole-unique-sequence-based analysis (WUSA).** PDRPL/NGS data were produced by biopanning of preimmune (PI), non-protective (P14, Y14, and Z14), and protective (P28a, P28b, Y28, and Z28) rabbit serum samples. Blue lines drawn across the PCA plot (A) separate the 3 groups of serum samples.

samples. Out of these 3 peptides, GLLQKPL was most discriminating as it had the highest positive difference in the number of its sequencing reads between the PR and NP samples (Fig 2B). Consistently, the PCA analysis showed a high degree of data randomness with the first principle component capturing only 21% of the variability. Despite this randomness, the PI, NP, and PR samples were still segregated by the PCA plots (Fig 4A). Interestingly, the respective heatmap did not discriminate between the PI and NP samples (Fig 4B). Consistent with the MA method results, the duplicate samples, P28a and P28b, were also clustered together; although this time this overlap was slightly higher (Fig 4B).

## Improving on the MA and WUSA methods

The high degree of data randomness shown by both WUSA and MA methods may be due to a high level of noise inherently associated with the PDRPL/NGS. In addition, the observed randomness could indicate that only a small fraction of antibody repertoires of infected NZW rabbits is related to an anti-*B. burgdorferi* immune response; whereas the majority of rabbit antibody repertoires may simply reflect the previous exposure to unrelated antigens. To increase the signal to noise ratio, this time we generated the PCA plots and respective heatmaps only for the first 25 motifs and the first 25 7-mer sequences as ranked by the *t*-test. The PCA plots showed that the overall separation between the PR and NP samples were noticeably improved by both WUSA and MA methods (Fig 5A and 5C). Overall, the heatmap generated via the MA method reflected the immunological status (protective vs. non-protective responses) more accurately than the WUSA-based heatmap (Fig 5B and 5D). Curiously, the arrangement of the 8 samples from left to right showed the progression of protective immune response against the LD pathogen. Furthermore, sample Z28 from the NZW rabbit that had developed its protective antibodies later than the other animals was placed on the border between the PR and NP samples and was adjacent to sample Z14 derived from the same rabbit (Fig 5B). The replicates, P28a and P28b, were clustered together and the PI sample was distantly positioned from the 4 PR samples (Fig 5B). On the contrary, the heatmap generated via

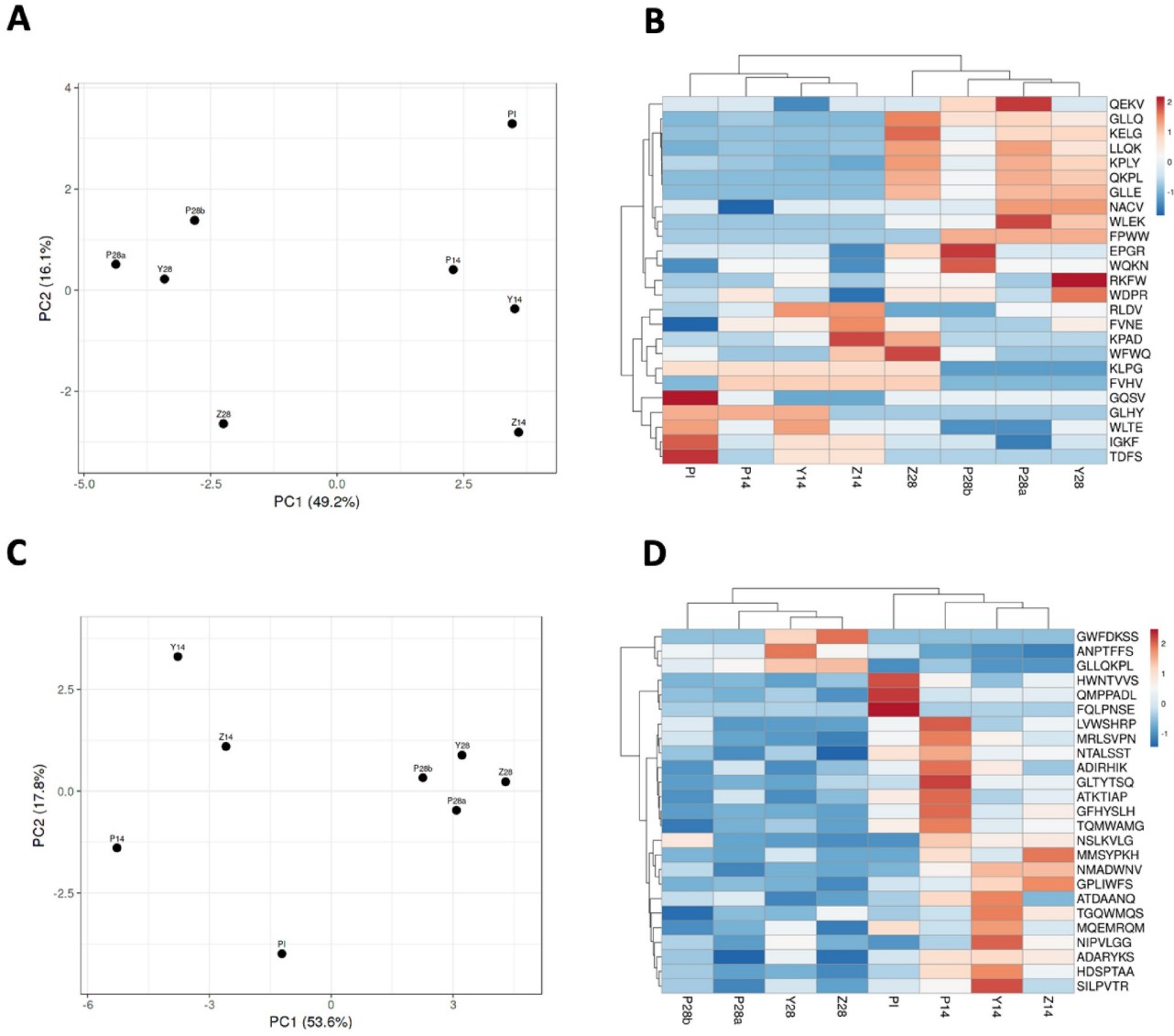

**Fig 5. ClustVis-generated Principal Component Analysis plots (A and C) and heatmaps (B and D) produced for the first 25 most statistically significant 4-mer motifs (A and B) and unique 7-mer sequences (C and D).** PDRPL/NGS data were generated by biopanning of preimmune (PI), non-protective (P14, Y14, and Z14), and protective (P28a, P28b, Y28, and Z28) rabbit serum samples.

the WUSA, placed the PI sample between the PR and NP samples; and yet the technical replicates, P28a and P28b, were consitently grouped together (Fig 5D).

## Discussion

Reverse vaccinology is a powerful vaccine development approach, which involves initial identification of vaccine targets via genome-based computational analysis [92]. In our recent studies, we utilized a subtractive reverse vaccinology to delineate putative surface epitopes of *B. burgdorferi* associated with antibody-mediated protection against the LD pathogen [10, 32, 93]. By using the two LD animal models, protective and non-protective immune sera from C3H mice and NZW rabbits were analyzed via the PDRPL/NGS and compared within each animal species [10, 93]. To further evaluate the utility of the PDRPL/NGS for quantitative analysis of

immune responses, in the present study, we conveniently made use of the PDRPL/NGS datasets previously generated from the NZW rabbit model [93].

Although the PDRPL/NGS is a very sensitive high throughput method for identifying disease-specific mimotopes recognized by serum antibodies [94]; its reproducibility and accuracy are low [30] compared to the random peptide array [14, 15, 95]. There are at least two probable causes for the low reproducibility. First, the binding of low adundant antibodies to their cognate targets is a stochastic process, since, during the first round of selection, these targets are present in very low numbers. Second, there is a loss of relevant sequences over secondary rounds of selection and propagation of phages [30]. Importantly, these multiple rounds of selection and amplification are required for making the ratio of a target to its cognate antibody large enough to reach the positive correlation between the number of sequencing reads for a given unique peptide and a titer of its specific antibody. However, because of these required secondary rounds, the phage amplification becomes highly sensitive to fluctuations in the number of bacterial cells successfully infected by competing phage particles. As a result, the numbers of sequencing reads in technical replicates may show a large variation, which in turn reduces the overall accuracy of the quantitative analysis of the PDRPL/NGS data.

To improve the reproducibility of the PDRPL/NGS method, in the present study, we evaluated the feasibility of using 4-mer motifs as potential biomarkers of the protective anti-*B. burgdorderi* immune response. Given that most of the binding energy is typically derived from 4–6 amino acids [17], the specificity of antigen-antibody interaction is defined by only a few anchor residues within an epitope [29, 96–98]. Previouusly, we [27] and others [34] demonstrated that the four-amino-acid consensus is a minimum motif length, which is sufficient to define an antibody specificity. Thus, the high copy number of shorter targets in the initial selection makes the process of target recognition more deterministic. While the unique 7-mer sequence is represented on average by almost 80 copies, the tetramer is represented by a much greater number of copies: $10^{11}/20^4 = 625,000$. Moreover, it was previously noted that the size of a peptide family (a total number of sequence-related peptides) sharing a given motif positively correlated with a number of sequencing reads for the most represented sequence of that family and the titer of serum antibodies recognizing that motif [29]. Importantly, this positive correlation was experimentally confirmed via ELISA by utilizing immobilized motif-containing peptides [29]. Thus, considering the numbers of distinct motif-constituting sequences transforms qualitative peptide differences into a quantitative parameter, which more accurately reflects titers of target-specific antibodies.

The present study showed that, in general, the MA method discriminated between the protective and non-protective serum samples as equally well as the WUSA with a couple of noteworthy differences. First, although the number of the discriminating 7-mer sequences identified by the *t*-test ($\alpha = 0.1$) is a little higher than that of discriminating motifs (11 vs. 9), only 3 out of 11 whole sequence peptides were represented by higher numbers of their respective sequencing reads for the PR samples compared to the NP samples. In contrast, 7 out of 9 motifs well discriminated the PR samples ($p = 0.024619$ by Chi-Square test). Second, the unsupervised hierarchical clustering obtained via the MA method arranged the samples in the order that more accurately reflected the development of the protective anti-*B. burgdorferi* immune response compared to the WUSA-based clustering. The observed outcome difference can be attributed to the fact that the MA method is more quantitative and, hence, more discriminating by nature compared to the WUSA approach. For example, when group A and group B sera (3 samples in each group) are compared and each of the 3 samples of group A has only 1 unique 7-mer; and together these 7-mers form a motif, than it is only the motif (rather than the 7-mers) that will disciriminate between the two groups. Also, the MA method tends to level off high fluctuations in numbers of sequencing reads for 7-mer sequences. For example,

despite the GLLEKLH peptide was consistently detected and absent in all the PR and NP samples, respectively; the high variance in its sequencing reads prevented this 7-mer from distinguishing between the two serum groups. To contrast, the GLLE motif shared by this and other 7-mers of the 4 PR samples was well discriminating. Therefore, the MA approach allows each of motif-constituting 7-mers to quantitatively contribute to the overall discriminatory power of the respective motif regardless of high variations in its sequencing reads. Given that a total number of a given unique peptide positively correlates with the number of its respective sequencing reads [29]; motifs, in general, are likely to more accurately reflect both quantitative and qualitative alterations in antibody repertoires.

## Methods

### Bacteria and rabbit infection

In the prior study [32], *B. burgdorferi* B31-A3 strain [99] was utilized to infect New Zealand White (NZW) rabbits. Spirochetes were cultivated in liquid Barbour-Stoenner-Kelly medium supplemented with 6% rabbit serum (Gemini Bio-Products, USA; referred to here as BSK-II) and incubated at 35˚C under 2% $CO_2$. For our prior study [32], a total of 3 female NZW rabbits of 12–14 weeks of age (rabbits P, Y, and Z; 3.0–3.5 kg by weight each) were purchased from Charles River Laboratories (USA). The rabbits were singly housed in an animal BSL-2 facility at Texas A&M University. The animals were daily subjected to visual welfare-related assessment for the entire duration of their stay. In order to obtain preimmune sera, the 3 rabbits were bled via the marginal ear vein after their acclimation. Each animal was then individually inoculated intradermally at six sites along the spine with *B. burgdorferi* B31-A3 strain at $10^6$ cells per each inoculation site as described [88]. The infection of each animal was verified by culture-positive skin biopsies taken around inoculation sites (3 mm in diameter) at weeks 1, 2, and 3 postinfection. Skin tissues collected at week 4 postinfection were also collected and showed that all the skin biopsies were culture-negative with an exception being NZW rabbit Z, whose skin culture was weakly positive. In addition to very low numbers of spirochetes in the NZW rabbit Z skin culture, spirochetes exhibited no or very little (impaired) motility, which indicated the final stage of spirochetal clearance [32]. The blood samples were placed at 4˚C until next day. Serum samples were then generated from each blood sample via centrifugation at 5,000 x g for 10 minutes. The serum samples were stored at -80˚C. To prevent bacterial and fungal contamination during skin biopsy culture, BSK-II was supplemented with 0.02 mg ml$^{-1}$ phosphomycin, 0.05 mg ml$^{-1}$ rifampicin, and 2.5 mg ml$^{-1}$ amphotericin B. All the cultures were weekly examined for 6 weeks via dark-field microscopy for the presence of viable spirochetes. At week 4 postinfection, the NZW 3 rabbits were humanely sacrificed by applying isoflurane.

### Phage display (Ph.D.) library

Twenty μl of each rabbit serum sample and 10 μl of random peptide library Ph.D.-7 (NEB, MA, USA) were diluted in 200 μl of TRIS Buffered Saline buffer with 0.1% Tween 20 (referred to here as TBST) and 1% bovine serum albumin and incubated at 25˚C for 18 hr [29]. Antibody-bound phages were isolated by utilizing protein G-agarose beads (Santa Cruz Biotechnology, Inc., TX, USA) to the phage antibody mixture for 1 hr. The bead mixture was transferred to a 96-well MultiScreen-Mesh filter plate (EMD Millipore, MA, USA) with a 20-μm-pore-size nylon mesh at the bottom. To remove unbound phages, vacuum was used to the exterior of the nylon mesh. The beads were washed 4 times with 100 μl of TBST buffer for each well. Antibody-bound phages were eluted with 100 μl of 100 mM Tris-glycine buffer (pH 2.2) and subsequently neutralized by adding 20 μl of 1 M Tris buffer (pH 9.1). The solution was

then used for amplification of eluted phages by infecting bacteria per the manufacturer's instructions. Amplified phages were subjected to the next round of biopanning, after which antibody-bound phages were isolated by using protein G-agarose beads. After washing the beads in the 96-well MultiScreen-Mesh filter plate, DNA was isolated by phenol-chloroform extraction and ethanol precipitation. Lastly, 21-nucleotide-long DNA fragments that encoded random peptides were amplified by PCR utilizing the following forward and reverse primers, respectively: 5′–AATGATACGGCGACCACCGAGATCTACACTCTTTCCCTA CACGACGCTCTTCCGAT CT(INDEX)TGGTACCTTTCTATTCTCACTCT–3′ and 5′– CAAGCAGAAGAGGGCATACGAGCTCTTCCGATCTAACAGTTTCGGCCGAACCTCCACC–3′. The INDEX of each forward primer was defined by the unique six-nucleotide long sequence. Thus, for each serum, a forward primer with the unique INDEX was used [29]. After purification, generated PCR-amplified DNA library was multiplexed and sequenced by 50 single read cycles via Illumina HiSeq 2500 platform at University of Buffalo Genomics and Bioinformatics Core, New York State Center of Excellence Bioinformatics & Life Sciences, Buffalo, New York.

### DNA reads analysis

The sequencing resulted in approximately $1.79 \times 10^8$ DNA reads. Reads were de-multiplexed based on the unique bar codes. Each read was tagged by the unique INDEX and contained a 21- nucleotide-long sequence, which encoded a given peptide. The 21-nucleotide sequences were extracted between positions 30 and 50 and translated into 7-mers in the first frame. The mean number of peptides that had no stop codon per serum was $5.9 \times 10^6$, of which approximately $2.1 \times 10^5$ peptides per sample were non-redundant. The numbers of peptides were normalized in order that the total number of peptides to be equal between samples and the first 1,000 most abundant unique peptide sequences were selected for each serum samples. For each serum sample, S3 Table shows the first 1,000 unique peptides with the corresponding normalized number of sequencing reads for each unique peptide sequence.

### Generating motifs using MEME software

One thousand most abundant unique peptide sequences were uploaded to MEME software available online (http://meme-suite.org/tools/meme). The parameters of running the algorithm were selected to generate 50 motifs. To make the 7-mer sequences compatible with the MEME tool requirements (the minimum length is 8 letters), the letter X was added to each 7-mer. Prior to running the program, we chose that each motif has the minimum sites of 3 (to unify at least 3 unique sequences) and the width of 4 amino acids exactly.

### Declarations

### Ethics approval and consent to participate

NZW rabbits were maintained at Texas A&M University in an animal facility accredited by the Association for the Assessment and Accreditation of Laboratory Animal Care International. All the experimental practices, which involved animals, had been approved by the Institutional Animal Care and Use Committee of Texas A&M University (IACUC 2017–0390) and were carried out in accordance with Public Health Service Policy on Humane Care and Use of Laboratory Animals (2002), Guide for the Care and Use of Agricultural Animals in Research and Teaching (2010), and Guide for the Care and Use of Laboratory Animals (2011).

## Supporting information

**S1 Table. Sheet 1.** Frequency distribution for the discriminating 4-mer motifs with *p*-values calculated by the *t*-test to distinguish between the non-protective (P14, Y14, and Z14) and protective (P28, Y28, and Z28) rabbit serum samples. **Sheet 2.** Frequency distribution for the discriminating 4-mer motifs that were ranked by the difference between the minimum values of the protective serum samples (P28, Y28, and Z28) and the maximum values of the non-protective serum samples (P14, Y14, and Z14). Highlighted in yellow are statistically significant motifs.
(XLSX)

**S2 Table. Sheet 1**. Frequency distribution for the discriminating unique 7-mer sequences with *p*-values calculated via the *t*-test to distinguish between the non-protective (P14, Y14, and Z14) and protective (P28, Y28, and Z28) rabbit serum samples. **Sheet 2.** Frequency distribution for the discriminating unique 7-mer sequences that were ranked by the difference between the minimum values of the protective serum samples (P28, Y28, and Z28) and the maximum values of the non-protective serum samples (P14, Y14, and Z14).
(XLSX)

**S3 Table. The first one thousand of the most abundant 7-mer peptides with the respective numbers of their sequencing reads for each rabbit serum sample.** PI denotes pooled preimmune serum sample. P14, Y14, and Z14 are non-protective serum samples. P281, P28b, Y28, and Z28 are protective serum samples. P28a and P28b are technical replicates.
(XLSX)

## Author Contributions

**Conceptualization:** Yurij Ionov, Artem S. Rogovskyy.

**Data curation:** Yurij Ionov, Artem S. Rogovskyy.

**Formal analysis:** Yurij Ionov, Artem S. Rogovskyy.

**Funding acquisition:** Artem S. Rogovskyy.

**Investigation:** Yurij Ionov, Artem S. Rogovskyy.

**Methodology:** Yurij Ionov.

**Project administration:** Artem S. Rogovskyy.

**Resources:** Artem S. Rogovskyy.

**Supervision:** Artem S. Rogovskyy.

**Validation:** Artem S. Rogovskyy.

**Visualization:** Artem S. Rogovskyy.

**Writing – original draft:** Yurij Ionov.

**Writing – review & editing:** Artem S. Rogovskyy.

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
