## [Decision Letter · Decision Letter 0]

20 Sep 2019

PONE-D-19-21468

Comparison of motif-based and unique sequence-based analyses of phage display library datasets generated by biopanning of anti-Borrelia burgdorferi immune sera

PLOS ONE

Dear Dr Ionov,

Thank you for submitting your manuscript to PLOS ONE. After careful consideration, we feel that it has merit but does not fully meet PLOS ONE’s publication criteria as it currently stands. Therefore, we invite you to submit a revised version of the manuscript that addresses the points raised during the review process.

All three reviewers identified technical issues and inaccuracies in the text that need to be addressed and attended to as part of the revision.

We would appreciate receiving your revised manuscript by Nov 04 2019 11:59PM. To enhance the reproducibility of your results, we recommend that if applicable you deposit your laboratory protocols in protocols.io, where a protocol can be assigned its own identifier (DOI) such that it can be cited independently in the future. For instructions see: http://journals.plos.org/plosone/s/submission-guidelines#loc-laboratory-protocols

We look forward to receiving your revised manuscript.

Kind regards,

Nicholas J Mantis

Academic Editor

PLOS ONE

Journal Requirements:

Reviewers' comments:

Reviewer's Responses to Questions

**Comments to the Author**

1. Is the manuscript technically sound, and do the data support the conclusions?

Reviewer #1: Partly

Reviewer #2: Yes

Reviewer #3: Partly

2. Has the statistical analysis been performed appropriately and rigorously? 

Reviewer #1: No

Reviewer #2: Yes

Reviewer #3: Yes

3. Have the authors made all data underlying the findings in their manuscript fully available?

Reviewer #1: Yes

Reviewer #2: Yes

Reviewer #3: Yes

4. Is the manuscript presented in an intelligible fashion and written in standard English?

Reviewer #1: Yes

Reviewer #2: Yes

Reviewer #3: Yes

5. Review Comments to the Author

Reviewer #1: This manuscript by Ionov and Rogovskyy uses a phage display system, coupled with Next Generation sequencing, to identify epitopes or mimotopes of polyclonal serum antibodies against the spirochete Borrelia burgdorferi. They obtain sequencing counts of full-length peptides, or motifs found within them, and compare the results between protective and non-protective serum of New Zealand rabbits. They determine that both methods (full length and motif) provide discriminating power between protective and non-protective serum, but that the motif-based method has some advantages, which they discuss. While the manuscript is interesting and could provide new methods of Lyme diagnosis and/or post vaccine protection determination, there are several issues that should be addressed before acceptance.

Major Issues

1) The math in the paragraph between lines 91 and 107 should be checked, and also described in more detail. In some cases the math appears wrong, and in other cases it’s unclear where a particular number comes from. For example:

a. On line 95, it states that in 10 uL of library, each 7-mer is represented by 10^11 phages, and is thus represented 70 times. This is likely supposed to say that in 10 uL there are 10^11 phages, and since there are 20^7 (not 7^20 as stated) possible 7-mers, that averages to ~78 of each 7-mer in the sample. The authors may also want to state that the library documentation states that the phage concentration is 10^13 phages per mL, which is where the 10^11 phages in 10 uL number comes from.

b. The next sentence makes a point that there are 6.4x10^7 possible 6-mer sequences, and thus the library is big enough to represent them all. But the previous sentence has already stated that the library is big enough to not only contain all possible 7-mer sequences, but to do so ~80 times each.

c. On line 102, where does the “2” in “2x10^11” come from? Previously it was established that the 10 uL sample of library contains 10^11 phages, not 2x10^11.

d. Please also explain where the rest of the numbers in this paragraph (32,000, 4x10^11, 1.6x10^5) come from, or how they are calculated.

2) On lines 137-140, it states that you are identifying motifs of 4 amino acids, and clustering peptides that contain that motif that differ by “only few” amino acids. How can the 4-mer motif still be considered intact if a “few” amino acids are different? Is this instead referring to the other 3 amino acids in the 7-mer peptide that aren’t part of the motif? Either way, please clarify.

3) On line 198, Figure 1 is referenced, and the text says that it shows the most abundant motif for sample Z28. Figure 1 shows the motif QKPL, but in Supplementary Table 2, the most abundant motif for sample Z28 appears to be KIGD. QKPL is not even the most abundant motif in sample Z28 among the motifs deemed to be statistically significant.

4) I’m not a statistician, but I think there are a few issues with the way the statistics were performed on the sample sets. First, on line 205 it says that significance testing was done between the “three non-protective and four protective sera.” Indeed, looking at the provided supplemental tables in Excel, that is how the calculation was done. However, there were not four independent protective sera, only three. One of the protective serum samples, P28, was analyzed in duplicate. I don’t believe you can use BOTH of those data sets that came out of the P28 sample together in the t-test calculation, because they are not independent data sets. Second, it states on lines 203-204 that a “two sample two tailed t-test (assuming unequal variances)” was used. However, since these sets consist of serum samples that were taken from the same set of three animals at two different time points, shouldn’t the test have been a paired t-test? P14 and P28 are paired, as are Y14/Y28 and Z14/Z28. And finally, why was p<0.1 chosen as the cutoff for significance, when generally p<0.05 is used? Again, I don’t have a strong statistical background, so please do explain if I’m wrong about these issues.

5) On lines 233-238, there seems to be a contradiction. It first states that you identified 13 distinct 7-mers that could discriminate between protective and non-protective serum, and you note that these 13 are identified by having p<0.1 (not 0.01), meaning that they are significant. But it then says that out of these 13 peptides, only GLLQKPL was significantly different between protective and non-protective serum.

6) Please explain Figures 3 and 4 in clearer terms. In the text, the only thing it says about Figures 3A and 4A is that a tool was used to separate the eight sera samples by principal component analysis. However, it does not explain what the two principal components are, or what the graph axes mean (what are the percentages?). Similarly, Figures 3B and 4B don’t really explain how to interpret them. For instance, in both heat maps the duplicate samples (P28a and P28b) do not match very well, with red and blue colors not matching up at all. I would think that these samples should look almost identical, right? It also says that the preimmune serum is well segregated from the other samples, but it doesn’t look to me any more different from the other samples than P28a looks from P28b.

Minor Issues

1) Comparing lines 128-129 with lines 165-166, you have two different things that are abbreviated as “PI” (post-infection and preimmune). One is labeled as lower case pi, while the other is uppercase PI, but I do think the letters should be different between the two abbreviations as well.

2) On line 165, one of the day-14 samples is listed as Z28, instead of Z14.

3) On line 235, p<0.01 should be p<0.1.

4) In the methods on line 321, it says that skin tissues that were collected at week 4 post infection show all culture-negative biopsies, but earlier, on line 221, it states that the day 28 skin biopsy of animal Z was still culture-positive.

5) The Ethics paragraph starting on line 383 is largely identical to the paragraph starting on line 404.

Reviewer #2: This manuscript describes the use of two methods employing phage displayed random peptide libraries coupled to Next Gen Sequencing (PDRPL/NGS) to examine antibody target epitopes in the context of Borrelia burgdorferi (Bb) infection. The authors use motif-based and unique sequence-based analyses of phage display library datasets to map the protective and non-protective antibody target epitopes in rabbits infected with Bb.

1. The study is based on the premise that New Zealand rabbits can clear the Bb infection solely because of their protective antibodies (as opposed to humans). However, not enough evidence is provided to directly support this premise. For example, is it clear that the rabbit antibodies against the identified epitopes are more protective than human antibodies against Bb?

2. The manuscript does not seem to link the identified epitopes to any parent proteins in borrelia. This would be important and helpful information to include and discuss.

3. The introduction discusses the advantages of PDRPL/NGS, but doesn’t really mention the potential disadvantages and biases of the PDRPL/NGS approach in epitope mapping.

4. The are some inaccurate statements concerning Lyme disease in the manuscript. For example, that “LD human patients, who have not been timely treated with antimicrobials [46-51], remain infected with the spirochete for life despite very strong anti-B. burgdorferi antibody responses”. In fact, the majority of Lyme patients’ infection resolves spontaneously even without antibiotics (refer to papers from A. Steere and others from the 80s and 90s). The paper can benefit greatly from further careful review to make sure the text is accurate and well-referenced with regard to statements about Lyme disease.

5. The manuscript can also benefit from further editing for typos, grammar, and structure. There are many long and convoluted sentences that seriously reduce the readability of the paper.

Reviewer #3: The submitted manuscript "Comparison of motif-based and unique sequence-based analyses of phage display library datasets generated by biopanning of anti-Borrelia burgdorferi immune sera" is very interesting both from the theoretical and practical point of view. I recommend proceeding with publication process after minor revision.

There is an only minor correction which should be addressed

Minor revisions, recommendations

Line 75: protective epitopes – authors should stick to protection-associated epitopes. Protective epitope should be isolated and proved in vivo.

Line 154-155: Short notice describing both approaches in a few words should be included. There is mention about “the first approach” the second one follows much later but the text in between concerns both of them.

Line 400: TBST is Tris-buffered with a tween. TBST is well-known abbreviations; moreover, it is explained in Material and Methods. There is no need to include it into the list.

The discussion should be rephrased more extensively

In discussion data obtained should be compared to other publications. It is not a summary of the results. There are some ideas what might be discussed or authors can find better topics:

The methods used can be compared to other available data or previously published methods. The strengths and weaknesses might be described. This would also be a good place to try to find out the identity of protein containing the epitope GLLQkpl if possible and to describe properties essential for “protectiveness” – physiological function, expression pattern, accessibility for antibodies and so on. Or it would be possible to highlight other epitopes revealed by analysis even with lower significance especially when they might be mapped to known surface proteins.

6. PLOS authors have the option to publish the peer review history of their article (what does this mean?). If published, this will include your full peer review and any attached files.

Reviewer #1: No

Reviewer #2: No

Reviewer #3: No

---

## [Author Response · Author response to Decision Letter 0]

11 Oct 2019

Response to the reviewers

PONE-D-19-21468

Comparison of motif-based and unique sequence-based analyses of phage display library datasets generated by biopanning of anti-Borrelia burgdorferi immune sera

PLOS ONE

Dear Nicholas J Mantis,

First we want express our gratitude to the very helpful criticism and commentaries of the reviewers that enabled us to essentially improve our manuscript. Below are our responses indicated in bold to each point raised by the three reviewers. 

Please do not hesitate to contact me directly, if any more questions arise.

Sincerely,

Yurij Ionov, PhD

Journal Requirements:

2. Please include captions for your Supporting Information files at the end of your manuscript, and update any in-text citations to match accordingly. Please see our Supporting Information guidelines for more information:http://journals.plos.org/plosone/s/supporting-information.

The captions for the Supporting Information files have been included at the end of the manuscript and the citations have been updated. 

Responses to Reviewer #1: 

This manuscript by Ionov and Rogovskyy uses a phage display system, coupled with Next Generation sequencing, to identify epitopes or mimotopes of polyclonal serum antibodies against the spirochete Borrelia burgdorferi. They obtain sequencing counts of full-length peptides, or motifs found within them, and compare the results between protective and non-protective serum of New Zealand rabbits. They determine that both methods (full length and motif) provide discriminating power between protective and non-protective serum, but that the motif-based method has some advantages, which they discuss. While the manuscript is interesting and could provide new methods of Lyme diagnosis and/or post vaccine protection determination, there are several issues that should be addressed before acceptance.

Major Issues

1) The math in the paragraph between lines 91 and 107 should be checked, and also described in more detail. In some cases the math appears wrong, and in other cases it’s unclear where a particular number comes from. For example:

a. On line 95, it states that in 10 uL of library, each 7-mer is represented by 10^11 phages, and is thus represented 70 times. This is likely supposed to say that in 10 uL there are 10^11 phages, and since there are 20^7 (not 7^20 as stated) possible 7-mers, that averages to ~78 of each 7-mer in the sample. The authors may also want to state that the library documentation states that the phage concentration is 10^13 phages per mL, which is where the 10^11 phages in 10 uL number comes from. 

The math in the indicated part of the text was corrected and described in more details to clarify the undertaken calculations. 

b. The next sentence makes a point that there are 6.4x10^7 possible 6-mer sequences, and thus the library is big enough to represent them all. But the previous sentence has already stated that the library is big enough to not only contain all possible 7-mer sequences, but to do so ~80 times each.

The text was corrected to eliminate the possible redundancies. 

c. On line 102, where does the “2” in “2x10^11” come from? Previously it was established that the 10 uL sample of library contains 10^11 phages, not 2x10^11.

The “2” comes from the fact that each 7-mer contains two distinct 6-mers starting from the first and the second position of the 7-mer. 

d. Please also explain where the rest of the numbers in this paragraph (32,000, 4x10^11, 1.6x10^5) come from, or how they are calculated

All calculations were corrected and explained in the revised text..

2) On lines 137-140, it states that you are identifying motifs of 4 amino acids, and clustering peptides that contain that motif that differ by “only few” amino acids. How can the 4-mer motif still be considered intact if a “few” amino acids are different? Is this instead referring to the other 3 amino acids in the 7-mer peptide that aren’t part of the motif? Either way, please clarify.

We included the clarification in the text why we used the phrase “few amino acids” but not exactly the 3 amino acids. This is because each letter in the motif can be the combination of distinct letters as can be seen in the Figure1. The numbers for the motifs were generated by MEME software and not by our counting of particular tetramers in all samples.

3) On line 198, Figure 1 is referenced, and the text says that it shows the most abundant motif for sample Z28. Figure 1 shows the motif QKPL, but in Supplementary Table 2, the most abundant motif for sample Z28 appears to be KIGD. QKPL is not even the most abundant motif in sample Z28 among the motifs deemed to be statistically significant.

We corrected this error in the revised text.

4) I’m not a statistician, but I think there are a few issues with the way the statistics were performed on the sample sets. First, on line 205 it says that significance testing was done between the “three non-protective and four protective sera.” Indeed, looking at the provided supplemental tables in Excel, that is how the calculation was done. However, there were not four independent protective sera, only three. One of the protective serum samples, P28, was analyzed in duplicate. I don’t believe you can use BOTH of those data sets that came out of the P28 sample together in the t-test calculation, because they are not independent data sets. Second, it states on lines 203-204 that a “two sample two tailed t-test (assuming unequal variances)” was used. However, since these sets consist of serum samples that were taken from the same set of three animals at two different time points, shouldn’t the test have been a paired t-test? P14 and P28 are paired, as are Y14/Y28 and Z14/Z28. And finally, why was p<0.1 chosen as the cutoff for significance, when generally p<0.05 is used? Again, I don’t have a strong statistical background, so please do explain if I’m wrong about these issues.

Although the reviewer modestly claims that he is not an expert in statistics his comments were quite enlightening for us. We corrected our calculations and used statistics according to his recommendations. For example we used paired t-test instead oftwo-tailed t-test with unequal variances, and we removed one replica from duplicate protective sample to recalculate the p-values. Although p< 0.05 is generally used the significance level P<0.1 is also acceptable. From Google: “The significance level is denoted by α and is the probability of rejecting the null hypothesis, if it is true. Typical values for α are 0.01, 0.05 and 0.1. It is a value that we select based on the certainty we need.”

5) On lines 233-238, there seems to be a contradiction. It first states that you identified 13 distinct 7-mers that could discriminate between protective and non-protective serum, and you note that these 13 are identified by having p<0.1 (not 0.01), meaning that they are significant. But it then says that out of these 13 peptides, only GLLQKPL was significantly different between protective and non-protective serum.

We corrected this contradiction in the text that was the consequence of the poor wording. We really had in mind that this peptide was visually distinct between the protective and non-protective sera on the figure. 

6) Please explain Figures 3 and 4 in clearer terms. In the text, the only thing it says about Figures 3A and 4A is that a tool was used to separate the eight sera samples by principal component analysis. However, it does not explain what the two principal components are, or what the graph axes mean (what are the percentages?). Similarly, Figures 3B and 4B don’t really explain how to interpret them. For instance, in both heat maps the duplicate samples (P28a and P28b) do not match very well, with red and blue colors not matching up at all. I would think that these samples should look almost identical, right? It also says that the preimmune serum is well segregated from the other samples, but it doesn’t look to me any more different from the other samples than P28a looks from P28b.

 We provided better explanations of the graphs and plots in the revised text. Poor matching of the blue and red is caused by inherent high noise of the method, however, despite the high noise the math clusters the samples correctly in the motif based approach. 

Minor Issues

1) Comparing lines 128-129 with lines 165-166, you have two different things that are abbreviated as “PI” (post-infection and preimmune). One is labeled as lower case pi, while the other is uppercase PI, but I do think the letters should be different between the two abbreviations as well.

We removed “pi” as abbreviation for “postinfection” for the better clarity. 

2) On line 165, one of the day-14 samples is listed as Z28, instead of Z14.

This has been corrected. Thank you for catching it.

3) On line 235, p<0.01 should be p<0.1.

We corrected the p values accordingly.

4) In the methods on line 321, it says that skin tissues that were collected at week 4 post infection show all culture-negative biopsies, but earlier, on line 221, it states that the day 28 skin biopsy of animal Z was still culture-positive.

Thank you for pointing this out. This has been corrected and clarified accordingly. Please see the text. 

5) The Ethics paragraph starting on line 383 is largely identical to the paragraph starting on line 404.

The Ethics paragraph starting on line 383 was removed. Thank you for pointing this out.

Reviewer #2: 

This manuscript describes the use of two methods employing phage displayed random peptide libraries coupled to Next Gen Sequencing (PDRPL/NGS) to examine antibody target epitopes in the context of Borrelia burgdorferi (Bb) infection. The authors use motif-based and unique sequence-based analyses of phage display library datasets to map the protective and non-protective antibody target epitopes in rabbits infected with Bb.

1. The study is based on the premise that New Zealand rabbits can clear the Bb infection solely because of their protective antibodies (as opposed to humans). However, not enough evidence is provided to directly support this premise. For example, is it clear that the rabbit antibodies against the identified epitopes are more protective than human antibodies against Bb?

We agree with the reviewer and now we are contrasting the ability of NZW rabbit immune sera (not human sera) to prevent B. burgdorferi infection to anti-Borrelia mouse sera. Please see the respective change in the text.

2. The manuscript does not seem to link the identified epitopes to any parent proteins in borrelia. This would be important and helpful information to include and discuss.

We fully agree with the reviewer that it is a logical next step of the work. However, given 1) that the focus of this manuscript is solely to compare the two methods in identifying discriminatory epitopes between the protective and non-protective sera; 2) that we used only limited number of peptides (n=1,000) rather than the all available peptides for this comparative analysis; 3) that mapping of identified epitopes/motifs is an extensive analysis in itself (due to the fact that mapping has to be performed to the entire proteome of B. burgdorferi); and 4) that we plan to do this mapping study by using the methods described in this work and utilizing the full data sets for the next manuscript; we would like to refrain from discussing this aspect in the current manuscript. 

In addition it is known that only about 10% of all antibodies recognize the linear epitopes that can be identified by sequence alignment. The majority of antibodies are against conformational epitopes or even against carbohydrate structures on the surface of cells and such identification is simply impossible. The identified peptides represent the mimotopes and if they are protective they can be used for vaccination without knowledge of their real target. 

3. The introduction discusses the advantages of PDRPL/NGS, but doesn’t really mention the potential disadvantages and biases of the PDRPL/NGS approach in epitope mapping.

We expanded our discussion and included the comparing with the alternative method. 

4. There are some inaccurate statements concerning Lyme disease in the manuscript. For example, that “LD human patients, who have not been timely treated with antimicrobials [46-51], remain infected with the spirochete for life despite very strong anti-B. burgdorferi antibody responses”. In fact, the majority of Lyme patients’ infection resolves spontaneously even without antibiotics (refer to papers from A. Steere and others from the 80s and 90s). The paper can benefit greatly from further careful review to make sure the text is accurate and well-referenced with regard to statements about Lyme disease.

Although there is some evidence that B. burgdorferi can still persist in humans with Lyme disease symptoms despite antibiotic treatment (e.g., Middelveen MJ, Sapi E, Burke J, Filush KR, Franco A, Fesler MC, Stricker RB: Persistent Borrelia Infection in Patients with Ongoing Symptoms of Lyme Disease. Healthcare (Basel) 2018, 6(2)), we still see with the reviewer argument and hence rephrased the respective statement in order to avoid this highly controversial topic among the readers. 

5. The manuscript can also benefit from further editing for typos, grammar, and structure. There are many long and convoluted sentences that seriously reduce the readability of the paper.

We are sorry to hear that. Accordingly we worked on improving the readability of this manuscript. 

Reviewer #3: 

The submitted manuscript "Comparison of motif-based and unique sequence-based analyses of phage display library datasets generated by biopanning of anti-Borrelia burgdorferi immune sera" is very interesting both from the theoretical and practical point of view. I recommend proceeding with publication process after minor revision.

There is an only minor correction which should be addressed

Minor revisions, recommendations

Line 75: protective epitopes – authors should stick to protection-associated epitopes. Protective epitope should be isolated and proved in vivo.

This has been corrected. Thank you for pointing this out.

Line 154-155: Short notice describing both approaches in a few words should be included. There is mention about “the first approach” the second one follows much later but the text in between concerns both of them.

We corrected the text according to this recommendation. 

Line 400: TBST is Tris-buffered with a tween. TBST is well-known abbreviations; moreover, it is explained in Material and Methods. There is no need to include it into the list.

Per your request, the abbreviation “TBST” was removed from the list.

The discussion should be rephrased more extensively

In discussion data obtained should be compared to other publications. It is not a summary of the results. There are some ideas what might be discussed or authors can find better topics: The methods used can be compared to other available data or previously published methods. The strengths and weaknesses might be described. 

We corrected the revised text according to this recommendation 

This would also be a good place to try to find out the identity of protein containing the epitope GLLQkpl if possible and to describe properties essential for “protectiveness” – physiological function, expression pattern, accessibility for antibodies and so on. Or it would be possible to highlight other epitopes revealed by analysis even with lower significance especially when they might be mapped to known surface proteins.

We fully agree with the reviewer that it is a logical next step of the work. However, given 1) that the focus of this manuscript is solely to compare the two methods in identifying discriminatory epitopes between the protective and non-protective sera; 2) that we used only limited number of peptides (n=1,000) rather than the all available peptides for this comparative analysis; 3) that mapping of identified epitopes/motifs is an extensive analysis in itself (due to the fact that mapping has to be performed to the entire proteome of B. burgdorferi); and 4) that we plan to do this mapping study by using the methods described in this work and utilizing the full data sets for the next manuscript; we would like to refrain from discussing this aspect in the current manuscript. Besides, the epitope GLLQKPL may be the mimotope of the conformational epitope consisting of amino acids brought together on the surface of a protein by the pattern of protein folding

---

## [Editor Report · Decision Letter 1]

16 Oct 2019

PONE-D-19-21468R1

Comparison of motif-based and unique sequence-based analyses of phage display library datasets generated by biopanning of anti-Borrelia burgdorferi immune sera

PLOS ONE

Dear Dr Ionov,

Thank you for submitting your manuscript to PLOS ONE. After careful consideration, we feel that it has merit but does not fully meet PLOS ONE’s publication criteria as it currently stands. Therefore, we invite you to submit a revised version of the manuscript that addresses the points raised during the review process.

As you requested in an email to me,  I am returning the manuscript to you for revisions to the Supplemental Tables before resubmission. 

We would appreciate receiving your revised manuscript by Nov 30 2019 11:59PM. To enhance the reproducibility of your results, we recommend that if applicable you deposit your laboratory protocols in protocols.io, where a protocol can be assigned its own identifier (DOI) such that it can be cited independently in the future. For instructions see: http://journals.plos.org/plosone/s/submission-guidelines#loc-laboratory-protocols

We look forward to receiving your revised manuscript.

Kind regards,

Nicholas J Mantis

Academic Editor

PLOS ONE

---

## [Author Response · Author response to Decision Letter 1]

16 Oct 2019

Response to the reviewers

PONE-D-19-21468

Comparison of motif-based and unique sequence-based analyses of phage display library datasets generated by biopanning of anti-Borrelia burgdorferi immune sera

PLOS ONE

Dear Nicholas J Mantis,

First we want express our gratitude to the very helpful criticism and commentaries of the reviewers that enabled us to essentially improve our manuscript. Below are our responses indicated in bold to each point raised by the three reviewers. 

Please do not hesitate to contact me directly, if any more questions arise.

Sincerely,

Yurij Ionov, PhD

Journal Requirements:

2. Please include captions for your Supporting Information files at the end of your manuscript, and update any in-text citations to match accordingly. Please see our Supporting Information guidelines for more information:http://journals.plos.org/plosone/s/supporting-information.

The captions for the Supporting Information files have been included at the end of the manuscript and the citations have been updated. 

 

Responses to Reviewer #1: 

This manuscript by Ionov and Rogovskyy uses a phage display system, coupled with Next Generation sequencing, to identify epitopes or mimotopes of polyclonal serum antibodies against the spirochete Borrelia burgdorferi. They obtain sequencing counts of full-length peptides, or motifs found within them, and compare the results between protective and non-protective serum of New Zealand rabbits. They determine that both methods (full length and motif) provide discriminating power between protective and non-protective serum, but that the motif-based method has some advantages, which they discuss. While the manuscript is interesting and could provide new methods of Lyme diagnosis and/or post vaccine protection determination, there are several issues that should be addressed before acceptance.

Major Issues

1) The math in the paragraph between lines 91 and 107 should be checked, and also described in more detail. In some cases the math appears wrong, and in other cases it’s unclear where a particular number comes from. For example:

a. On line 95, it states that in 10 uL of library, each 7-mer is represented by 10^11 phages, and is thus represented 70 times. This is likely supposed to say that in 10 uL there are 10^11 phages, and since there are 20^7 (not 7^20 as stated) possible 7-mers, that averages to ~78 of each 7-mer in the sample. The authors may also want to state that the library documentation states that the phage concentration is 10^13 phages per mL, which is where the 10^11 phages in 10 uL number comes from. 

The math in the indicated part of the text was corrected and described in more details to clarify the undertaken calculations. 

b. The next sentence makes a point that there are 6.4x10^7 possible 6-mer sequences, and thus the library is big enough to represent them all. But the previous sentence has already stated that the library is big enough to not only contain all possible 7-mer sequences, but to do so ~80 times each.

The text was corrected to eliminate the possible redundancies. 

c. On line 102, where does the “2” in “2x10^11” come from? Previously it was established that the 10 uL sample of library contains 10^11 phages, not 2x10^11.

The “2” comes from the fact that each 7-mer contains two distinct 6-mers starting from the first and the second position of the 7-mer. 

d. Please also explain where the rest of the numbers in this paragraph (32,000, 4x10^11, 1.6x10^5) come from, or how they are calculated

All calculations were corrected and explained in the revised text.

2) On lines 137-140, it states that you are identifying motifs of 4 amino acids, and clustering peptides that contain that motif that differ by “only few” amino acids. How can the 4-mer motif still be considered intact if a “few” amino acids are different? Is this instead referring to the other 3 amino acids in the 7-mer peptide that aren’t part of the motif? Either way, please clarify.

We included the clarification in the text why we used the phrase “few amino acids” and not exactly the 3 amino acids. Each letter in the motif can be a combination of distinct letters as can be seen in Figure1. The numbers for the motifs were automatically generated by MEME software.

3) On line 198, Figure 1 is referenced, and the text says that it shows the most abundant motif for sample Z28. Figure 1 shows the motif QKPL, but in Supplementary Table 2, the most abundant motif for sample Z28 appears to be KIGD. QKPL is not even the most abundant motif in sample Z28 among the motifs deemed to be statistically significant.

We corrected this error in the revised text.

4) I’m not a statistician, but I think there are a few issues with the way the statistics were performed on the sample sets. First, on line 205 it says that significance testing was done between the “three non-protective and four protective sera.” Indeed, looking at the provided supplemental tables in Excel, that is how the calculation was done. However, there were not four independent protective sera, only three. One of the protective serum samples, P28, was analyzed in duplicate. I don’t believe you can use BOTH of those data sets that came out of the P28 sample together in the t-test calculation, because they are not independent data sets. Second, it states on lines 203-204 that a “two sample two tailed t-test (assuming unequal variances)” was used. However, since these sets consist of serum samples that were taken from the same set of three animals at two different time points, shouldn’t the test have been a paired t-test? P14 and P28 are paired, as are Y14/Y28 and Z14/Z28. And finally, why was p<0.1 chosen as the cutoff for significance, when generally p<0.05 is used? Again, I don’t have a strong statistical background, so please do explain if I’m wrong about these issues.

Although the reviewer modestly claims that he is not an expert in statistics his comments were quite enlightening for us. We corrected our calculations and used statistics according to his recommendations. For example, we used the paired t-test instead of the two-tailed t-test with unequal variances, and we removed one replicated sample of the protective sample to recalculate the p-values. Although p< 0.05 is generally used the significance level p<0.1 is also acceptable.

5) On lines 233-238, there seems to be a contradiction. It first states that you identified 13 distinct 7-mers that could discriminate between protective and non-protective serum, and you note that these 13 are identified by having p<0.1 (not 0.01), meaning that they are significant. But it then says that out of these 13 peptides, only GLLQKPL was significantly different between protective and non-protective serum.

We corrected this. What we had in mind that is this peptide was visually distinct between the protective and non-protective sera on the figure. 

6) Please explain Figures 3 and 4 in clearer terms. In the text, the only thing it says about Figures 3A and 4A is that a tool was used to separate the eight sera samples by principal component analysis. However, it does not explain what the two principal components are, or what the graph axes mean (what are the percentages?). Similarly, Figures 3B and 4B don’t really explain how to interpret them. For instance, in both heat maps the duplicate samples (P28a and P28b) do not match very well, with red and blue colors not matching up at all. I would think that these samples should look almost identical, right? It also says that the preimmune serum is well segregated from the other samples, but it doesn’t look to me any more different from the other samples than P28a looks from P28b.

We provided better explanations of the graphs and plots in the revised text. Poor matching of the blue and red is caused by inherent high noise of the method, however, despite the high noise, the samples are correctly clustered in the motif-based approach. 

Minor Issues

1) Comparing lines 128-129 with lines 165-166, you have two different things that are abbreviated as “PI” (post-infection and preimmune). One is labeled as lower case pi, while the other is uppercase PI, but I do think the letters should be different between the two abbreviations as well.

We removed “pi” as abbreviation for “postinfection” for the better clarity. 

2) On line 165, one of the day-14 samples is listed as Z28, instead of Z14.

This has been corrected. Thank you for catching it.

3) On line 235, p<0.01 should be p<0.1.

We corrected the p-values accordingly.

4) In the methods on line 321, it says that skin tissues that were collected at week 4 post infection show all culture-negative biopsies, but earlier, on line 221, it states that the day 28 skin biopsy of animal Z was still culture-positive.

Thank you for pointing this out. This has been corrected and clarified accordingly. Please see the text. 

5) The Ethics paragraph starting on line 383 is largely identical to the paragraph starting on line 404.

The Ethics paragraph starting on line 383 was removed. Thank you for pointing this out.

Responses to Reviewer #2: 

This manuscript describes the use of two methods employing phage displayed random peptide libraries coupled to Next Gen Sequencing (PDRPL/NGS) to examine antibody target epitopes in the context of Borrelia burgdorferi (Bb) infection. The authors use motif-based and unique sequence-based analyses of phage display library datasets to map the protective and non-protective antibody target epitopes in rabbits infected with Bb.

1. The study is based on the premise that New Zealand rabbits can clear the Bb infection solely because of their protective antibodies (as opposed to humans). However, not enough evidence is provided to directly support this premise. For example, is it clear that the rabbit antibodies against the identified epitopes are more protective than human antibodies against Bb?

We agree with the reviewer and now we are contrasting the ability of NZW rabbit immune sera (not human sera) to prevent B. burgdorferi infection to anti-Borrelia mouse sera. Please see the respective change in the text.

2. The manuscript does not seem to link the identified epitopes to any parent proteins in borrelia. This would be important and helpful information to include and discuss.

We fully agree with the reviewer that it is a logical next step of the work. However, given 1) that the focus of this manuscript is solely to compare the two methods in identifying discriminatory epitopes between the protective and non-protective sera; 2) that we used only limited number of peptides (n=1,000) rather than the all available peptides for this comparative analysis; 3) that mapping of identified epitopes/motifs is an extensive analysis in itself (due to the fact that mapping has to be performed to the entire proteome of B. burgdorferi); and 4) that we plan to do this mapping study by using the methods described in this work and utilizing the full data sets for the next manuscript; we would like to refrain from discussing this aspect in the current manuscript. 

Finally, it is known that only about 10% of all antibodies recognize the linear epitopes that can be identified by sequence alignment. The majority of antibodies are against conformational epitopes or even against carbohydrate structures on the surface of cells and such identification is simply impossible. The identified peptides represent the mimotopes and if they are protective they can be used for vaccination without knowledge of their real target. 

3. The introduction discusses the advantages of PDRPL/NGS, but doesn’t really mention the potential disadvantages and biases of the PDRPL/NGS approach in epitope mapping.

We expanded our discussion and included the comparing with the alternative method. 

4. There are some inaccurate statements concerning Lyme disease in the manuscript. For example, that “LD human patients, who have not been timely treated with antimicrobials [46-51], remain infected with the spirochete for life despite very strong anti-B. burgdorferi antibody responses”. In fact, the majority of Lyme patients’ infection resolves spontaneously even without antibiotics (refer to papers from A. Steere and others from the 80s and 90s). The paper can benefit greatly from further careful review to make sure the text is accurate and well-referenced with regard to statements about Lyme disease.

Although there is some evidence that B. burgdorferi can still persist in humans with Lyme disease symptoms even after antibiotic treatment (e.g., Middelveen MJ, Sapi E, Burke J, Filush KR, Franco A, Fesler MC, Stricker RB: Persistent Borrelia Infection in Patients with Ongoing Symptoms of Lyme Disease. Healthcare (Basel) 2018, 6(2)), we still see with the reviewer argument and hence rephrased the respective statement in order to avoid this highly controversial topic among the readers. 

5. The manuscript can also benefit from further editing for typos, grammar, and structure. There are many long and convoluted sentences that seriously reduce the readability of the paper.

We are sorry to hear that. Accordingly we worked on improving the readability of this manuscript. 

Responses to Reviewer #3: 

The submitted manuscript "Comparison of motif-based and unique sequence-based analyses of phage display library datasets generated by biopanning of anti-Borrelia burgdorferi immune sera" is very interesting both from the theoretical and practical point of view. I recommend proceeding with publication process after minor revision.

There is an only minor correction which should be addressed

Minor revisions, recommendations

Line 75: protective epitopes – authors should stick to protection-associated epitopes. Protective epitope should be isolated and proved in vivo.

This has been corrected. Thank you for pointing this out.

Line 154-155: Short notice describing both approaches in a few words should be included. There is mention about “the first approach” the second one follows much later but the text in between concerns both of them.

We corrected the text according to this recommendation. 

Line 400: TBST is Tris-buffered with a tween. TBST is well-known abbreviations; moreover, it is explained in Material and Methods. There is no need to include it into the list.

Per your request, the abbreviation “TBST” was removed from the list.

The discussion should be rephrased more extensively

In discussion data obtained should be compared to other publications. It is not a summary of the results. There are some ideas what might be discussed or authors can find better topics: The methods used can be compared to other available data or previously published methods. The strengths and weaknesses might be described. 

We corrected the revised text according to this recommendation 

This would also be a good place to try to find out the identity of protein containing the epitope GLLQkpl if possible and to describe properties essential for “protectiveness” – physiological function, expression pattern, accessibility for antibodies and so on. Or it would be possible to highlight other epitopes revealed by analysis even with lower significance especially when they might be mapped to known surface proteins.

We fully agree with the reviewer that it is a logical next step of the work. However, given 1) that the focus of this manuscript is solely to compare the two methods in identifying discriminatory epitopes between the protective and non-protective sera; 2) that we used only limited number of peptides (n=1,000) rather than the all available peptides for this comparative analysis; 3) that mapping of identified epitopes/motifs is an extensive analysis in itself (due to the fact that mapping has to be performed to the entire proteome of B. burgdorferi); and 4) that we plan to do this mapping study by using the methods described in this work and utilizing the full data sets for the next manuscript; we would like to refrain from discussing this aspect in the current manuscript. Finally, the epitope GLLQKPL may be the mimotope of the conformational epitope consisting of amino acids brought together on the surface of a protein by the pattern of protein folding.

---

## [Decision Letter · Decision Letter 2]

26 Nov 2019

Comparison of motif-based and whole-unique-sequence-based analyses of phage display library datasets generated by biopanning of anti-Borrelia burgdorferi immune sera

PONE-D-19-21468R2

Dear Dr. Ionov,

We are pleased to inform you that your manuscript has been judged scientifically suitable for publication and will be formally accepted for publication once it complies with all outstanding technical requirements.

With kind regards,

Nicholas J Mantis

Academic Editor

PLOS ONE

Additional Editor Comments (optional):

Reviewers' comments:

Reviewer's Responses to Questions

**Comments to the Author**

1. If the authors have adequately addressed your comments raised in a previous round of review and you feel that this manuscript is now acceptable for publication, you may indicate that here to bypass the “Comments to the Author” section, enter your conflict of interest statement in the “Confidential to Editor” section, and submit your "Accept" recommendation.

Reviewer #2: All comments have been addressed

2. Is the manuscript technically sound, and do the data support the conclusions?

Reviewer #2: Yes

3. Has the statistical analysis been performed appropriately and rigorously? 

Reviewer #2: Yes

4. Have the authors made all data underlying the findings in their manuscript fully available?

Reviewer #2: Yes

5. Is the manuscript presented in an intelligible fashion and written in standard English?

Reviewer #2: Yes

6. Review Comments to the Author

Reviewer #2: The authors have addressed the critiques from this reviewer.

The authors have addressed the critiques from this reviewer.

7. PLOS authors have the option to publish the peer review history of their article (what does this mean?). If published, this will include your full peer review and any attached files.

Reviewer #2: Yes: Armin Alaedini

---

## [Editor Report · Acceptance letter]

3 Dec 2019

PONE-D-19-21468R2 

Comparison of motif-based and whole-unique-sequence-based analyses of phage display library datasets generated by biopanning of anti-*Borrelia burgdorferi* immune sera 

Dear Dr. Ionov:

I am pleased to inform you that your manuscript has been deemed suitable for publication in PLOS ONE. Congratulations! Your manuscript is now with our production department. 

With kind regards,

on behalf of

Dr. Nicholas J Mantis 

Academic Editor

PLOS ONE